# The Past, the Present, and the Future of the Size Exclusion Chromatography in Extracellular Vesicles Separation

**DOI:** 10.3390/v13112272

**Published:** 2021-11-13

**Authors:** Hussein Kaddour, Malik Tranquille, Chioma M. Okeoma

**Affiliations:** 1Department of Pharmacological Sciences, Renaissance School of Medicine, Stony Brook University, Stony Brook, NY 11794, USA; malik.tranquille@stonybrook.edu; 2Regeneron Pharmaceuticals, Inc., Tarrytown, NY 10591, USA

**Keywords:** extracellular vesicles (EVs), exosomes, size exclusion chromatography (SEC), gradient size exclusion chromatography (gSEC), ion exchange chromatography (IEX), hybrid chromatography

## Abstract

Extracellular vesicles (EVs) are cell-derived membranous particles secreted by all cell types (including virus infected and uninfected cells) into the extracellular milieu. EVs carry, protect, and transport a wide array of bioactive cargoes to recipient/target cells. EVs regulate physiological and pathophysiological processes in recipient cells and are important in therapeutics/drug delivery. Despite these great attributes of EVs, an efficient protocol for EV separation from biofluids is lacking. Numerous techniques have been adapted for the separation of EVs with size exclusion chromatography (SEC)-based methods being the most promising. Here, we review the SEC protocols used for EV separation, and discuss opportunities for significant improvements, such as the development of novel particle purification liquid chromatography (PPLC) system capable of tandem purification and characterization of biological and synthetic particles with near-single vesicle resolution. Finally, we identify future perspectives and current issues to make PPLC a tool capable of providing a unified, automated, adaptable, yet simple and affordable particle separation resource.

## 1. Introduction

EVs are intercellular communication vehicles that are revolutionizing the biomedicine field. However, a few people know that the discovery of EVs can be traced back to the 19th century, although under different nomenclature. In the 1840s, Gulliver’s editorial report (based on the work of Hewson in the 1770s) on milky particles in the blood serum ––that he called “the molecular base of the chyle” and characterized as very small globules of active Brownian movement and of uniform size ranging from ~0.5 to 1 micron––is perhaps the first encounter of EVs in the literature (see [1], pages 7 and 13). In the 1870s, Edmunds [2] described the normal blood serum as filled with a “nebulous haze of points, as is mote-laden air in a sunbeam” when visualized under a microscope. This finding was reproduced by Muller, who coined the particles “Haemokonien" (or blood dust) [3]. However, it was not until the 1930s when Frazer and Stewart reported on different subpopulations of particles in the blood serum, which they described as "large bright particles, small bright particles, and dull particles [4]. The authors also reported particle abundance during fasting and after meals, and estimated particle size to be ~1, ~0.33–0.5, and 0.035 micron, respectively for the above-mentioned particles [5]. In 1946, Chargaff and West [6] isolated the coagulation components of the platelet-free plasma by high-speed centrifugation (31,000× *g*), but it was not clear whether these structures were the same as *Haemokonien*. Other names describing blood serum particles such as protein particles, fat particles, fat dust, disintegrated platelets, chylomicrons, or white blood cells were also introduced [7]. In 1967, Wolf introduced the term “*platelet-dust*” to distinguish the platelet-secreted coagulant-material-carrying particles from chylomicrons. The “*platelet-dust*” were pelleted by high-speed centrifugation (134,211× *g*, 3 h) and characterized by electron microscopy and immunological techniques [8]. In the early 1980s, EVs were identified as structures of either membranous or endosomal origins exhibiting enzymatic activities. In the last two decades, EVs started gaining attention after the discovery of their pivotal roles in various physiological and pathological processes, but also in diagnostics and therapeutics (see [9,10,11] for excellent reviews on the subject).

Today, it is established that almost every cell type from the three kingdoms of life––Eukarya, Bacteria, and Archaea––produces EVs. Thus, EV research has become a subject of interest to scientists across numerous disciplines from basic science in chemistry, biology, and virology to applied science in diagnostics, pharmaceutics, and medicine. Such global interest in the EV field prompted a series of organizational actions in the past decade. The international Society for Extracellular Vesicles (ISEV) was created (2012 in Sweden) [12], and the *Journal of Extracellular Vesicles* (JEV, dedicated for EV research) was launched [13]. National EV societies across the world (United States—ASEMV, Spain—GEIVEX, Austria—ASEV, Germany—GSEV, Netherlands—NLSEV, Japan—JSEV, United Kingdom—UKEV, Taiwan—TSEV, Singapore—SOCRATES, Italy—EVITA, France—FSEV, Belgium—BESEV, and Korea—KSEV) were established, along with some regional networks (Regional Research Network on Extracellular Vesicles and the Gordon Research Conference Extracellular Vesicles biannual meeting). Furthermore, the number of scientific publications have exponentially increased, concomitant with an increase in industrial and governmental investment in EV research, such as the National Institute of Health (NIH) initiation of the Extracellular RNA consortium [14]. These measures were unsurprisingly accompanied with a massive increase in the global EV-based diagnostics and therapeutics market.

However, the progress and innovation in EV research has been largely impeded by challenges with EV separation and characterization [15]. Examples of such challenges include (1) insufficiently pure EV preparations; (2) interference of isolation reagents with downstream analyses; (3) lack of EV subpopulation separation; and (4) lack of scale-up capabilities. These challenges are exacerbated by the lack of automated equipment that readily isolate EVs from a given biofluid, the outcome of which is poor reproducibility of the EV preparations. Numerous NIH request for applications (RFA) have been activated to close this gap, but to date, no standardized EV separation method has been established. In addition to the lack of a standardized EV separation method, debate on whether single-vesicle resolution is needed for advancing the EV field is ongoing. Pro single-vesicle researchers argue that, because of the heterogeneity of EVs, identification of the specific bioactive EV cargo requires a single-vesicle approach. In contrast, researchers from another school of thought argue from a physiological standpoint that embracing rather than ignoring the inherent EV heterogeneity is key to understanding the biological and functional significance of EVs, since that is how EVs naturally occur in vivo. However, whether researchers are pro-single EV or not, it should be noted that before EVs can be efficiently used, the population(s) of interest must be isolated in pure and reproducible ways and retrieved in preparative quantities for downstream use. While searching for a consensus on standardized EV separation [16], the ISEV has published guidelines that urge EV scientists to thoroughly control for their EV content, use a combination of separation and characterization techniques, and report as much experimental details as possible [17,18,19,20,21].

As of 2016, ultracentrifugation was the most used technique for EV isolation, whether alone or in combination with other methods [22], despite its disadvantages [23]. Nevertheless, efforts concomitant with rapid technological advancement recently generated new exciting protocols for EV subpopulation separation (reviewed elsewhere [24]); e.g., asymmetric flow field-flow fractionation (AF4) [25], microfluidic isolation [26], high resolution flow cytometry [27,28,29,30,31], and immunocapture [32], to cite but a few. On the other hand, SEC has also been gaining popularity in the EV field, because of its simplicity, reproducibility, and gentle process allowing for purer and more active EVs, as compared to ultracentrifugation [33]. While EV separation techniques have been extensively discussed in various reviews [24,34] and protocols [35,36], this current review will focus on SEC and its variations as methods of EV separation. We will also discuss potential SEC add-ons that may help achieve EV subpopulation separation with near single-vesicle resolution.

## 2. History of Size Exclusion Chromatography (SEC) and Commercially Available SEC Beads

The first successful chemical separation by liquid-liquid partition chromatography can be traced back to the 1940s where Martin and Synge used a 1 cm silica gel column to separate amino acids [37,38,39]. For this innovation, the group was awarded the 1952 Nobel prize in chemistry [40]. However, the first size exclusion separation was described by Lindqvist and Storgårds who used a 2 cm × 120 cm starch column to separate cheese extracts into two peaks of peptides and amino-acids, collected in 275 fractions of 1 mL [41]. Soon after, glucan, a bacterial-derived carbohydrate polymer of glucose, was crosslinked with epichlorohydrin to yield crosslinked dextran, a more inert material with extended mechanical properties compared to starch and comparable fractionation capacity [42]. The capability of a dextran column was illustrated by separating 3 mL of fresh serum on a 17.5 cm × 2.5 cm into a large proteinaceous peak and small non-proteinaceous peak. By spiking the serum with a solution of saturated ammonium sulfate, the authors demonstrated the latter to be salts [42]. These dextran-based beads were commercialized at the time by (Pharmacia, Sweden) as Sephadex, and are still the most popular to date (Table 1). Ever since, minimal optimization was noted in which beads were rendered porous and the bead and pore sizes more controlled. Shortly after, Pharmacia commercialized Sepharose (agarose) and Sephacryl (polyacrylamide) beads, with a focus on improving mechanical strength to allow for high flow rates separation in fast purification liquid chromatography (FPLC) applications [43,44,45,46]. Variations of Sepharose were also developed for affinity purification [47,48], permitting fractionation of diverse bioactive molecules such as mammalian tRNA [49], human high density lipoproteins [50], and multiple types of growth factors in bone matrix [51]. Concurrently, Bio-Rad (CA, United States) commercialized Bio-Gel A (agarose) and P (polyacrylamide), originally prepared at the University of Uppsala (Sweden) by Hjertén and Mosbach [52,53] in 1962 for large and small molecules separation, respectively. Similar to Sephadex and Sepharose, the Bio-Gel products were also used in many discoveries [54,55,56,57,58], and are still on the market to date. Moreover, Tosoh, a Japanese chemical company, has recently extended its ToyoPEARL product line, a polymethacrylate-based resin mainly used for polymer purification with organic solvents, to include products for biological samples processing. A non-exhaustive list of commercially available SEC resins is presented in Table 1. Resins with an opportunity to isolate EV subpopulations are those with highest exclusion limit such as BioGel A-50 m, BioGelA-150 m, and Sephadex G-200. However, these products have been discontinued by their respective manufacturers.

## 3. Current SEC Methods for EV Separation

EVs is an umbrella term used to generically designate exosomes, ectosomes, microvesicles, microparticles, apoptotic bodies, oncosomes, exomeres, migrasomes, and many others [59,60]. EVs are present in cell culture fluids and bodily fluids and secretions, such as blood, saliva, tears, urine, cerebrospinal fluid (CSF), ascites, synovial fluid, amniotic fluid, semen, and breast milk [61,62,63,64]. EV are heterogeneous in structure and size, ranging from 30 nanometers up to 5 microns. This heterogeneity is driven by the type of the producer cell, its homeostatic condition, the type of biofluid, and the EV separation method. From a biophysical standpoint, the size of an EV is a critical parameter that defines the EV surface curvature, and thus dictates the surface cargo. For instance, the number of proteins that can physically fit on a large vesicle surface is orders of magnitude larger than that of a small vesicle (Figure 1). Similarly, large vesicles can carry more soluble cargo (i.e., nucleic acids and soluble proteins) in their intravesicular space, compared to smaller vesicles. In other words, the protein/lipid ratio (or nucleic acids/lipid ratio) is higher in large EVs versus small EVs. and may thus be an indication of EV bioactivity (Figure 1). This analysis is based on an assumption that a given EV population is homogeneous compositionally (i.e., secreted from a specific cellular subset), although in complex biofluids, size does not necessarily correlate with particle density. Nevertheless, the variation in EV size posits that EVs can theoretically be separated using size-guided techniques, such as SEC. In this section, we discussed the studies that used SEC for EV separation, grouped them into biofluid-type subsections, and highlighted the type of chromatographic methods used, in an effort to distill information that may facilitate in designing novel and improved SEC columns capable of effective EV separation. We will maintain the EV nomenclature used by authors in their publication.

### 3.1. SEC in Separating Cell Culture Medium

One of the first separations of cell-derived EVs using SEC columns is traced back to a 2012 study [65], where the authors used 16/60 Hiprep Sephacryl S400 HR column to purify exosomes secreted by mouse bone marrow-derived mesenchymal stem cells (MSCs) and MSCs from human umbilical cord Wharton’s jelly. Following separation, the authors showed that the anti-inflammatory activity of MSCs conditioned media (CM) is exosome-associated. Shortly after, two studies reported on the use of Superose 6 column for concentrated CM fractionation [66,67]. In the first study, the authors used a 50 cm column to separate astrocytes CM into four fractions, three of which were EVs [66]. In the second, a preparative Superose 6 column was used to isolate EVs from MCF-7, MDA-MB-231, L3.6pL, and U937 cells [67]. Interestingly, the representative separation profile that was shown [67], was very different from that of astrocytes [66], with only a small EV peak being detected in the void volume. Subsequently, Nordin et al. [68] reported higher yield and preservation of physical and functional properties of EVs isolated from conditioned stem cell media by ultra-filtration (UF)-SEC (S-400 Sephacryl column) compared to ultracentrifugation. Willms et al. [69] combined SEC on a S-1000 Sephacryl column to a 110,000 g ultracentrifugation pellet from 250 mL CM of B16F10 cells to show two compositionally different EV populations, large exosomes enriched for ACTN4 and CCNY and small exosomes enriched for EPHA2, although a complete exosome characterization that includes tetraspanins’ markers was not reported. It was also unclear what column was used because, in the Section 3, a HiPrep Sephacryl S-400 HR 16/60 column was reported, instead of a Sephacryl S-1000 [69]. In a separate study, SEC using a Sephacryl S-400 column outperformed ultracentrifugation when comparing the function of EVs isolated from the CM of 400 million CPC cells [70], but neither the EV chromatogram nor the SEC experimental details were reported [70]. Nonetheless, the same conclusion was reached by another group who cultured BEAS-2B cells and fractionated their CM on a 10 mL Sepharose CL-4B column after an ultrafiltration step (10 KDa), as opposed to the ultracentrifugation method (117,734× *g* for 2.5 h) [71]. Another size exclusion based technique that involves tangential flow filtration coupled with bind-elute SEC (TFF/BE-SEC) was developed and validated for the CM from two mouse cell lines, neuroblastoma N2a and myoblast C2C12 [72]. Today, pre-packed columns that can be used with or without FPLC with straightforward protocols are commercially available [73], although various cell culture conditions and parameters still need to be empirically determined.

### 3.2. Blood

One of the first successful EV separations from blood using SEC was reported in 2003, when Taylor et al. [74] isolated membrane fragments (MFs) from the sera of women with stage III ovarian cancer using a 1.5 cm × 4.5 cm bioGel A50m column. The authors found that compared to a healthy group, sera from ovarian cancer women contained significantly elevated levels of MFs. Fas Ligand (FasL), a type-II transmembrane protein that belongs to the tumor necrosis factor family, was associated with the tumor-derived MFs and was responsible for apoptosis of T lymphocytes, through loss of ζ-chain expression [74]. Taylor et al. also used Sepharose 2B column (1 cm × 35 cm) to isolate FasL-positive microvesicles that induced apoptosis of activated T lymphocytes from sera of oral cancer patients [75]. Later, the authors described a protocol for EV separation from blood plasma (as well as ascites) on a 2% agarose-based gel column (2.5 cm × 16 cm) for proteomics analysis and RNA profiling [76]. Based on their cumulative evidence in using SEC for isolating macromolecules, the authors concluded that use of precipitation reagents, such as ExoQuick, yielded increased quantity and quality of exosomal proteins and RNA, followed by SEC, then immunocapture; and the least effective method was ultracentrifugation [76]. However, as the science and technology of EV separation evolves, the precipitation method has been found to co-isolate EVs with other contaminants, and as such, is considered the least EV-specific method [77,78,79,80,81,82].

Arroyo et al. used Sephacryl S-500 to fractionate human plasma and showed that the majority of circulating miRNAs are associated with non-vesicular Ago-2 complexes, whereas only a few miRNA were EV-associated [83]. For instance, microRNAs miR-16 and miR-92a were mostly EV-free, whereas circulating let-7a was enriched in the EV fractions. On the other hand, by separating EVs from the High Density Lipoprotein (HDL) fractions using Superose 6 or Sephacryl S-200 FPLC columns, Vickers et al. showed that plasma miRNA are mostly HDL-associated [84]. In contrast, there exist studies demonstrating that extracellular RNAs are carried in EVs from human [85,86] and mouse [87] blood plasma. However, these studies did not use SEC for EV separation, making it difficult to compare.

Aside from commercially available SEC columns, the use of in-house columns has also been reported. Böing et al. [88] packed Sepharose CL-2B resin in a 10 mL plastic syringe stuffed with nylon stocking to create a 1.6 cm × 6.2 cm column. The column was used to demonstrate a single-step protocol for vesicles separation from human body fluids [88]. Soon after, this SEC method of EV separation from plasma was compared to that using a commercially available spin column (Exo-Spin^TM^) and, after detailed mass spectrometry analysis, the early SEC fractions, but not Exo-Spin derived EVs, were found to contain known EV markers [89]. This observation contradicts a concurrent report that concluded that Exo-Spin columns are excellent 10-min EV separation protocol from complex biofluid such as plasma [90]. The differences in these observations may be due to variations in donor EV cargo [62]. In their study, Baranyai et al. [91] used three different SEC columns (Sephacryl S-400, Sepharose 4B, Sepharose CL-2B) and compared the efficiency of blood plasma EVs separation to multiple rounds of ultracentrifugation. The authors showed that Sephacryl S-400 and Sepharose 4B, but not Sepharose CL-2B, efficiently separated EVs from albumin. It should be noted that EV mass ranges from mega to gigadaltons [92], and thus, several orders of magnitude larger than albumin which is 66.4 kDa. Thus, a mere separation between EV markers and albumin likely hides contaminants from other plasma particles with in-between molecular weight, such as HDL, LDL, and other immunoglobulins complexes. Nevertheless, Hong et al. [93] reported separation of intact and bioactive EVs from plasma of healthy controls and cancer patients using a mini-SEC 1.5 cm × 12 cm Sepharose 2B packed column. Subsequently, Gamez Valero et al. [94] compared this SEC method to the PEG precipitation method as well as the total protein organic solvent precipitation method from plasma samples to find that SEC outperformed the latter two methods, as evidenced by nanoparticle tracking analysis (NTA), flow cytometry, cryo-TEM, and immunoblot data. However, what is often ignored is that blood plasma is a complex fluid that contains multiple bionanoparticles of similar sizes and similar densities. Hence, it is likely very difficult to develop a one-step protocol that can efficiently separate all bionanoparticles from plasma [95], although this is still a matter of debate [96,97]. This limitation is highlighted in studies where plasma EVs were not separated from lipoproteins for proteomics studies, until a two-step protocol (density-gradient ultracentrifugation followed by SEC of the high-density fraction) was performed [98], this is also visible where studies that only used commercially available SEC columns co-purified non-EV particles with the EV preparations [99,100]. In any case, what is needed and not systematically performed is to test SEC fractions for other bionanoparticle markers, such as ApoB100, ApoA1, and Ago-2 proteins, in addition to tetraspanins and albumin. Another important control that is often overlooked is the inclusion of the non-EV fractions as a negative control for NTA, TEM, and immunoblot analyses.

### 3.3. Semen

Prostasomes are a class of EVs secreted by the prostate gland into the seminal plasma. The term “prostasomes” was coined in 1985 by Gunnar Ronquist (for more details about the history and function of prostasomes, see [101,102,103]). The chromatographic fractionation of seminal plasma is also decades old. Peeker et al. [104] fractionated clarified seminal plasma using a 1.6 cm × 60 cm Superdex 200 column, although their focus was not on prostasomes. On the other hand, Saez et al. [105] demonstrated an anti-oxidant property of human prostasomes that they purified using a combination of ultracentrifugation and Sephadex G-200 chromatography. More recently, Jones et al. [106] used SEC on a 35 mL Sephacryl S-300 column as an orthogonal method to demonstrate the presence of galectin-3, an immunomodulatory galactose-specific lectin, in human prostasomes. Two distinct prostasomes populations were isolated from seminal fluids from healthy vasectomized men using a combination of SEC on 70 cm × 2.6 cm Sephacryl S-1000 and density gradient ultracentrifugation [107,108]. The density-gradient ultracentrifugation step clearly separated a high-density luminal GLIPR2-rich population from a low-density membranous ANXA1-rich population [107]. Both populations exhibited atypical high levels of sphingomyelin, cholesterol, and glycosphingolipids and low levels of phosphatidylcholine, and differed from each other in the relative levels [108]. However, the chromatogram of the separation was not shown and it appears that the density gradient step is what allowed separation of seminal plasma into two different sizes of seminal EV (SEV)-populations, whereas SEC was used for washing EVs [108]. In their study, Lyu et al., used a 22 cm × 1 cm Sephadex G-50 column to isolate SEVs from HIV-infected and uninfected men who use or did not use illicit substances [109]. The separation profiles showed two peaks with the void peak containing the SEVs, as demonstrated by NTA, TEM, western blot, and turbidity data [109]. The group investigated the effects of HIV-1 infection and illicit drug use on the structure and function of SEVs, and concluded that HIV-SEV, Drug-SEV, and HIV-Drug-SEV may mediate pathological processes by differentially inducing monocytes’ actin-cytoskeletal re-organization, adhesion, ECM-modifying metalloprotease secretion, and chemotactic migration toward HIV secretome [109]. Lessons from study [109] led the group to develop a gradient SEC (gSEC) column in which all available Sephadex beads (Table 1) were layered in a 100 × 1 cm column [110]. The gSEC column yielded four distinct peaks, the first two of which correspond to two (large and small) EV populations, the third peak was enriched in membraneless condensates, comprising small RNAs and a unique protein signature, but clearly lacking a membrane whereas the fourth contained small molecules and salts [110]. Thus far, seminal plasma is one of the highest EV-containing body fluids, with diverse cargos and functions [62,111,112,113,114].

### 3.4. Urine

Urinary extracellular vesicles (uEV) have been proposed as source of biomarkers for kidney disorders, such as diabetic nephropathy, glomerular disease, and bladder cancer [115,116]. The clinical relevance of urine is not new and urinary proteins have been chromatographically fractionated for decades [117]. However, pathology-associated differences in uEV were not known until the study by Rood et al. [118]. The team used ultracentrifugation followed by SEC on a BioSep-SEC-S4000 column from Phenomenex (Torrance, CA, USA) to profile uEVs. The authors noted differences in chromatograms of urine from patients with membranous nephropathy or with focal segmental glomerulosclerosis compared to that of a healthy patient. Noteworthy is that the authors’ protocol involved multiple ultracentrifugation steps and protein reduction by DTT to remove soluble Tamm–Horsfall protein prior to SEC [118]. Interestingly, Kaddour et al. [110] obtained better resolved chromatogram of healthy urine using ultrafiltration followed by chromatography on a 100 cm × 1 cm Sephadex gSEC column, without ultracentrifugation or protein reduction. On the other hand, Lozano-Ramos [119] used a 10 mL Sepharose-CL-2B packed column to show clear EV separation from bulk urine proteins, as evidenced by NTA, flow-cytometry, cryo-TEM, immunoblot, and silver stain. The same group then performed in a separate study focusing on small RNA-Seq and proteomics analyses of uEVs from living and deceased kidney donors, but found only minor differences in the EV profiles that could not discriminate between the two groups [120]. Thus, questions on whether uEVs can be used as biomarkers for health and diseases remains largely open.

### 3.5. Other Bodily Fluids

SEC has been used to isolate EVs from other bodily fluids, including saliva, tears, milk, CSF, peritoneal dialysis efflux, and synovial fluid. Ogawa et al. [121] used 1.5 cm × 50 cm Sepharose CL-4B column to isolate saliva EVs containing active dipeptidyl peptidase IV (DPP IV) and galectin-3 and immunoglobulin A. Soon after, the same group used Sephacryl S-500 packed column (1.5 cm × 50 cm) to identify two saliva-derived EV populations [122]. Aqrawi et al. [123] used commercially available (Izon Science Ltd., New Zealand) qEV column to isolate EVs from saliva and tears. Sephacryl S-500 was also successfully used to fractionate milk fluff and serum into 4 and 5 peaks, respectively; allowing human and bovine milk EVs (MEVs) separation in a reproducible manner [124]. In their study, Kaddour et al. [110] used gSEC Sephadex column to obtain a 4-peak profile from cow milk, identifying a decreasing intensity in the EV peak in the order fat-free <2% fat < whole milk. Clarified CSF from young (<2 years) and old (>70 years) patients was purified on a S400-HR Sephacryl spin column and the filtrates were concentrated by the precipitation method (ExoQuick or total exosome isolation reagent) [125]. Although this method, has not been proven efficient for EV separation from any other biofluid, much less for CSF, the study identified over 50 differentially present miRNA between the two groups [125]. Sepharose CL-2B column was used post ultrafiltration as a concentrating step to isolate EVs from peritoneal dialysis (PD) efflux. A subsequent proteomics analysis revealed a good hierarchical clustering of samples from long term (>18 month) vs short-term (<10 month) PD patients [126]. Finally, separation of human synovial fluid (SF) EVs devoid of serum albumin and apolipoprotein A-I, was achieved on a HiPrep 26/60 Sephacryl S-500 HR column, outperforming ultracentrifugation and sucrose density-gradient ultracentrifugation [127]. Taken together, these studies demonstrate the superiority and usefulness of the SEC technique for separation of EVs from any biofluid type.

## 4. Discussion of the SEC Techniques Used for EV Separation

Particle size has been the key physical feature for EV definition; thus, a size-guided separation of EVs is unavoidable, potential crossover between subtypes notwithstanding. Yet, current size-exclusion columns, as reported in the section above, have been mostly used for bulk EV purification, with differing success. While our intent here was not to conduct a systematic review, it becomes evident from the 35 revisited studies that SEC is far from being standardized in the field, even within studies from the same group (Table 2). Furthermore, the use of automated systems such as HPLC and FPLC in the separation protocols are reported as automation is critical to overall performance and reproducibility, compared to the gravity-driven separations (Table 2). However, it becomes now evident that for heterogeneous bionanoparticle separation from complex biofluids, heterogeneous columns are needed, not only for separating EVs from non-EV components, but also for efficient one-step subpopulation separation. In this regard, gSEC columns coupled with a Particle Purification Liquid Chromatography (PPLC) system that can characterize EVs in real-time during separation would be valuable. The proof-of-concept of such approach was recently demonstrated by Kaddour et al., [110]. With empirical attention to separation parameters such as column length and width, sample volume, flow rate, column pressure, detectors parameters, and volume of collected fractions (see section below), it is possible to envisage a one-step protocol that achieves EV subpopulation separation with remarkable resolution.

## 5. Future Directions: Towards a Particle Purification Liquid Chromatography System

Despite the unanimous agreement in the field that SEC is a superior particle separation method [33,128], the method is still in need of empirical standardization with respect to the columns, the separation conditions, and the choice of the chromatographic system. We will discuss these parameters in this section. We will also highlight the benefits of a new EV purification system such as PPLC and how such a system may alleviate EVs separation challenges (Figure 2, Table 3).

### 5.1. Column Chemistry

In general, the column chemistry is of utmost importance in chromatographic separation. We have seen in Section 2 that resin manufacturing has been focused on bead homogeneity and decreased bead size, for increased reproducibility and accommodation of higher flow rates and faster runs. We have also seen a trend of resins with high exclusion limit discontinued from manufacturing. However, for the sought goals of EV subpopulation separation, both practices are amiss. Indeed, as noted previously EVs are in the mega- to gigadalton range [92], thus one would need beads with large exclusion limits (i.e, >500 MDa) and a gradient of different pore sizes [110] for a more resolved EV separation. An additional approach that may add significant resolution to EV separation is to employ columns of hybrid chemistries. In this regard, Van Deun et al. [129] showed that a dual-mode chromatography (DMC), employing ion exchange chromatography (IEC) and SEC column, can help isolate purer plasma EVs compared to SEC alone. While the proof of concept was validated, some basic but critical chromatographic principles were overlooked. For instance, the authors omitted an important control in which the IEC-bound fraction would be eluted and tested. It is possible that differently charged EV particles were eliminated as suggested by the decreased CD63 levels in DMC compared to SEC alone. In addition, the SEC resin was layered atop the IEC resin; but in this design, elution of the IEC-bound fraction would have not benefited from the presence of the SEC resin. In other words, the IEC resin played the role of a trap instead of a second dimension of separation. In contrast, if the IEC portion of the column were atop, the SEC portion of the column could have thus fractionated both IEC-unbound and IEC-bound material and operated as a desalting step for the bound-elution, resulting in a one-step 2D separation profile. Nevertheless, the results demonstrated a promising proof-of-concept that needs improvement. Another example of hybrid chromatography proof-of-concept is illustrated in an old study that used a PBS-washed 104k pellet of human seminal plasma in a three-step chromatographic purification: carboxymethyl cellulose (a cation exchange), Concanavalin A (a carbohydrate-binding protein), and 1.5 cm × 30 cm Sepharose CL-6B (size exclusion) to isolate and characterize an anti-fertility factor [130]. Although each of the separation steps was used independently and the resulting peaks were re-concentrated by ultracentrifugation before being applied to the next column, the successful separations support the concept of hybrid chromatography. In summary, available SEC columns and resins on the market do not provide enough resolution for EV subpopulation separation. However, there exist numerous opportunities for optimization, such as development of hybrid columns, that may significantly expand the dimensions for complex biofluid separation.

### 5.2. Online Characterization

Although SEC is often thought as a purification technique, it arguably offers a wide range of analytical potential [131] that is not well appreciated in the field. For instance, the molecular mass of a solute can mathematically be inferred, but also solute interactions and solute self-association can be characterized in terms of reaction stoichiometry and equilibrium constant [131]. These are valuable biochemical information. Furthermore, UV-Vis spectra can be employed to calculate biophysical information such as particle size and concentration. Strikingly however, the EV field has to date almost exclusively relied on analytical techniques that utilize light scattering detectors: DLS, NTA, and flow cytometry, known to have detection limits on the particle size, concentration determination, and labeling, respectively. There is paucity of studies in the literature that applied turbidity-based modeling for size determination of liposomes [132] and protocells [133]. However, Kaddour et al., [110] was the first to adapt Lorenz-Mie solution for EV size and concentration determination, taking advantage of HoloPy, a publicly available python package for digital holograms and light scattering analyses [134]. Moreover, SEC studies that used HPLC and FPLC systems often monitored elution profiles at wavelengths (220, 230, 260, or 280 nm) that confound EVs with non-EVs materials, such as extracellular condensates and other cell-free nucleic acids, proteins, and small molecules [110]. Since EVs are membranous and scatter light in the visible range, it is thus more relevant to monitor EVs in the turbidity range (400–600 nm). As other biological molecules may also exhibit absorbance around 400 nm, turbidity ratios R_1_ = A_400_/A_600_ and R_2_ = A_600_/A_650_ were defined to precisely distinguish between EVs and non-EVs during SEC separation [110].

The concept of online EV characterization during SEC is not limited to UV-Vis. Other sophisticated detectors can also be employed as long as they are non-invasive. In fact, DLS has been used for EV monitoring to precisely identify the EV-containing fractions when the EVs were overwhelmed by the presence of free-proteins in the UV range [135]. Fluorescence has also been employed for EV analysis during SEC separations and from which fluctuation correlation spectroscopy (FCS) analysis was employed to derive particle size [136]. However, DLS is beneficial only when EV are sufficiently diluted, unlike UV-Vis that can withstand relatively high EV concentrations. Additionally, DLS does not differentiate between EVs and possibly high scattering non-EVs particles such as HDL, LDL, and VLDL. On the other hand, fluorescence detection has great potential because, in addition to its capability to orthogonally determine vesicle size, antibody fluorescence can be used to determine levels of EV-associated tissue-specific markers in body fluids [136]. For instance, with a six-channel detector, it is possible to multiplex EV-positive, EV-negative, and tissue- or disease-specific markers, and obtain real-time information during the separation. This can be achieved in a one-step protocol, with minimal hands-on time, and without the need for a flow cytometer.

### 5.3. Fraction Collection

One of the fundamental limitations in the EV field arises from the lack of a high-resolution and reproducible EV sub-fractionation, exacerbating the EV inherent heterogeneity problem. What is also lacking is the retrieval of EV sub-fractions for subsequent functional studies, such as the determination of the protein and nucleic acid levels, as well as the deciphering of molecular and phenotypic effects of EVs on the target cells. Newly developed single-vesicle techniques (microfluidic devices, high-resolution confocal imaging, and nano-flow cytometry [137]) offer unique opportunities to tackle the formidable EV heterogeneity challenge, but leave the EV-retrieval problem unaddressed. The drawback of these techniques is that they require very diluted input material and/or are destructive in nature. In contrast, PPLC columns as described above, when coupled with a fast and small-volume fraction collector, have the potential of achieving EV sub-fractionation in near-native conditions and preparative amounts. Indeed, PPLC uses a multi-96 well plate fraction collector with as little as 20 microliters per fraction [110], whereas the majority of currently available FPLC and HPLC fraction collectors are designed for relatively large volume. Thus, the combination of high-resolution columns and small volume fraction collector of PPLC may allow retrieval of intact EV subpopulations in a convenient format for downstream compositional or functional studies.

### 5.4. Additional Factors for Consideration

#### 5.4.1. Pre and Post Column Concentrating Steps and Mobile Phase

Some of the samples such as cell culture supernatants and urine may require a pre-concentrating step before EV separation. Furthermore, depending on the downstream applications, purified EVs may also require re-concentration. The most used concentration method is ultrafiltration using membrane centrifugal devices ranging from 3 to 100 kDa. However, particles may irreversibly bind to the membrane filter leading to significant loss in yield. To overcome this hurdle, membrane pre-saturation with a 1% BSA solution and membrane wash with PBS post-concentrating are often recommended. However, when using a hybrid column (see Section 5.1 above), a pre-concentrating step of dilute input samples may not be needed anymore, because the ion exchange portion of the column can act as a pre-concentrating step. By contrast, hybrid IEX/SEC columns require longer runs with additional mobile phase solutions for a gradient- or step-elution of the IEX-bound material. Regarding the post-concentration step, an inexpensive and efficient alternative to ultrafiltration is concentration by reduced-pressure evaporation or by freeze-drying. In this case, the amount of salt in the sample should be noted and ideally, the mobile phase solution adapted (i.e., 0.1X PBS mobile phase in a 10X re-concentration protocol). In other cases, for instance when preparing samples for TEM analysis, 10 mM Tris buffer is more suitable than 1X PBS. However, changing buffers or lowering their concentration in mobile phase may allow non-specific binding between the sample proteins and the stationary phase. Thus, these separation parameters should be empirically determined and optimized per sample type.

#### 5.4.2. Scalability

As tissue-specific EVs circulating in some body fluids may be rare events compared to the other circulating materials, sample scale-up becomes unavoidable to obtaining preparative amounts of the specific EV subtype, needed for downstream applications. By contrast, other samples such as CSF and mosquito EVs [138] are very limited requiring rather a down-scale to accommodate low-volume input. Fortunately, PPLC, in particular, and chromatography, in general, are versatile. Indeed, chromatography separation utilizes various column sizes ranging from as little as a hundred microliters bead volume to over a hundred liters in industrial settings. Within a specific setting, chromatography systems are also versatile, by accommodating easily interchangeable tubing with various inner diameters. Finally, the PPLC fraction collection (see Section 5.3 above) can also be easily adapted to collect fractions of as little as 20 microliters up to 5 mm. The PPLC adaptability to the sample type and the needed scale, within the limits of the detectors at the given settings, is invaluable because it will permit freedom in experimental design for both analytical and preparative purposes.

#### 5.4.3. Biohazard Containment

There is need for systems that account for biohazard management, where separate columns/piping are used for separating infectious fluids and non-infectious fluids. For example, PPLC can be used for separating viruses from other EVs. Since PPLC is a closed system, it can be operated in a BSL1 laboratory. However, infectious fluids should be processed in a Class II BSC and in the BSL2+ laboratory, as adequate environment for handling fluids containing infectious materials, such as HIV. Furthermore, sodium azide (0.01–0.05%) may be added to the mobile phase to prevent bacterial contamination, especially when handling patient samples. However, if isolated EVs are meant for functional cell-culture based studies, sodium azide should be removed from the system.

## 6. Conclusions

In this review we have methodically provided summaries of primary data as evidence that SEC, and chromatography in general, can be leveraged to develop a multi-parameter resource for the purification, online characterization, and retrieval of preparative quantities of near-native, pure, and functional EV subpopulations from diverse biofluids. However, in spite of its long existence, affordability, and widespread availability, SEC is still in its infancy in the EV field. Hence, resources are needed for researchers and innovators to develop and standardize SEC-based EV separation protocols for each biofluid. Based on the EV separation techniques that are currently available, along with the potential of chromatography and recent improvements, such as PPLC, we envisage that the development of hybrid columns on a PPLC platform will be a compelling unifier for EV subpopulation separation and characterization. Such a hybrid system can also be applied to other biological vesicles such as viruses, or synthetic nanoparticles, including liposomes, polymers, and nanocages.

## 7. Patents

Stony Brook University filed a US patent application (PCT US2020/030914) covering the PPLC platform, with HK and CMO as inventors. Issues related to intellectual properties will be managed by the Office of Technology Licensing and Industry Relations (OTLIR) at Stony Brook University, according to existing and standard policies.

## Figures and Tables

**Figure 1 viruses-13-02272-f001:**
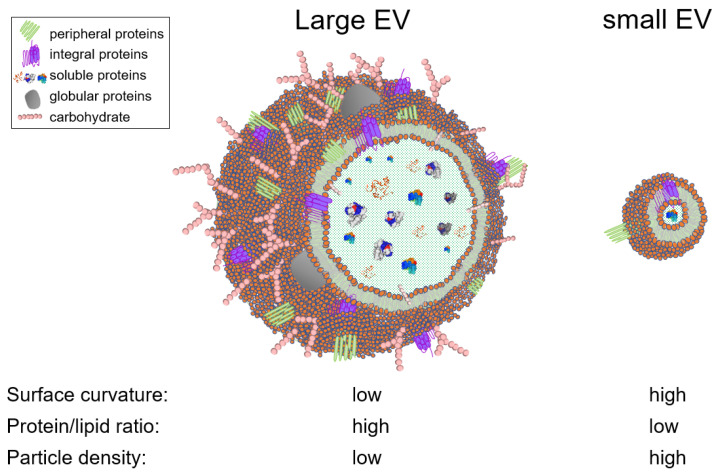
Schematic depicting the effects of the particle size on the composition and physiochemical properties of EVs. EVs contain other biomolecules that are not represented here, such as nucleic acids, small molecules.

**Figure 2 viruses-13-02272-f002:**
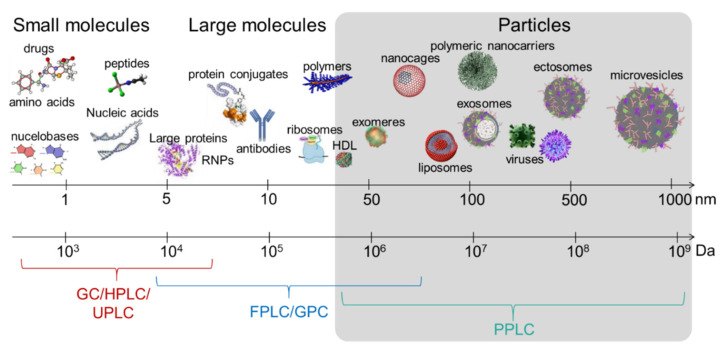
Schematic depicting the limits of the commercially available chromatographic systems in EV separation. GC, gas chromatography; HPLC, high performance liquid chromatography; UPLC, ultra-high performance liquid chromatography; FPLC, fast purification liquid chromatography; GPC, gel permeation chromatography. PPLC, particle purification liquid chromatography, which is currently under development (Kaddour et al.,), is tailored for mega- to gigadalton materials.

**Table 1 viruses-13-02272-t001:** Non-exhaustive list of commercially available SEC resins.

Company	Trade Name	Matrix	Product Number	Fractionation Range for Globular Protein (Da)
Cytvia ^¥^	Sephadex	Crosslinked dextran	G10	<700
			G15	<1500
			G25	1000–5000
			G50	1000–30,000
			G75	3000–80,000
			G100	4000–150,000
			G200 *	1000–200,000
	Sephacryl	Crosslinked allyl dextran and N,N’-methylene bisacrylamide	S-100 HR	1000–100,000
			S-200 HR	5000–250,000
			S-300 HR	10,000–1,500,000
			S-400 HR	20,000–8,000,000
			S-500 HR	40,000–20,000,000
	Sepharose	Crosslinked agarose	CL-2B	70,000–40,000,000
			CL-4B	70,000–20,000,000
			CL-6B	10,000–4,000,000
	Superose		6	5000–5,000,000
			12	1000–300,000
	Superdex	Composite dextran/crosslinked agarose	30	<10,000
			75	3000–70,000
			200	10,000–600,000
BioRad	Bio-Gel P	Crosslinked polyacrylamide and N,N’-methylene bisacrylamide	P-2	100–1800
			P-4	800–4000
			P-6	1000–6000
			P-10	1500–20,000
			P-30	2500–40,000
			P-60	3000–60,000
			P-100	50,000–100,000
	Bio-Gel A	Crosslinked agarose	A-0.5 m *	<10,000–500,000
			A-1.5 m *	<10,000–1,500,000
			A-5 m *	10,000–5,000,000
			A-15 m	40,000–15,000,000
			A-50 m *	100,000–50,000,000
			A-150 m *	100,000–150,000,000
Tosoh	ToyoPEARL	polymethacrylate	HW-40	>10,000 ^£^
			HW-50	>80,000 ^£^
			HW-55	>700,000 ^£^
			HW-65	>5,000,000 ^£^
			HW-75	>50,000,000 ^£^

^¥^ previously GE Healthcare, previously Pharmacia. * Sephadex G-200 and Bio-Gel A-0.5 m, A-1.5 m, A-5m, A-50 m, and A-150 m are discontinued by their respective manufacturer. ^£^ Exclusion limit.

**Table 2 viruses-13-02272-t002:** A non-exhaustive list of studies that used SEC for EV separation.

Study	Tissue/Cell Line/Biofluid	Column Size	Beads	Commercial/Lab Made	Fractions	System Used	Separation Profile Shown
Lee et al. [65]	Mouse bone marrow-derived MSCs, MSCs from human umbilical cord Wharton’s jelly, and human dermal fibroblasts	1.6 × 60 cm	Hiprep Sephacryl S-400 HR	prepacked (GE healthcare)	140 of 1 mL each	ÄKTA purifier liquid chromatography system	yes
Hajj et al. [66]	Astrocytes	50 cm	Superose 12 prep grade 3000-kDa exclusion	prepacked (GE healthcare)	25 of 3.5 mL each	NR	no
Redzic et al. [67]	cell lines: MCF-7, MDA-MB-231, L3.6pL and Hek293Fpl, U937	NR	preparatory Superose 6	prepacked (GE healthcare)	NR	NR	yes
Nordin et al. [68]	Cell lines: NSC-34, N2a, B16F10, HEK293T, iPSCs	NR	Sephacryl S-400 column	prepacked (GE healthcare)	2 mL	ÄKTA prime (GE Healthcare)	yes
Willms et al. [69]	Cell lines: B16F10, A431, N2a, H5V, hTERT	1.6 × 60 cm	HiPrep Sephacryl S-400 HR	prepacked (GE healthcare)	2 mL	ÄKTA prime (GE Healthcare)	yes
Mol et al. [70]	cell lines: CPCs and HMECs	NR	S400 highprep column	NR	NR	ÄKTAStart (GE Healthcare)	no
Benedikter et al. [71]	cell line: BEAS-2B	10 mL	Sepharose CL-4B	in-house	0.5 mL	gravity	no
Corso et al. [72]	N2a and myoblast C2C12		HiScreen Capto Core 700 column	GE Healthcare Life Sciences	NR	ÄKTA prime plus or ÄKTA Pure 25	yes
Taylor et al. [74]	human blood serum	1.5 × 45 cm	Bio-Gel A50 m	in-house	NR	NR	no
Kim et al. [75]	human blood serum, FasL-transduced PCI-13 cell lines and of normal human fibroblasts	1 × 35 cm	Sepharose 2B	in-house	1 mL	NR	no
Taylor et al. [76]	ascites specimens from women diagnosed with stage III serous adenocarcinoma of the ovary	2.5 × 16 cm	2% agarose (Agarose Bead Technologies, Tampa, FL)		2 mL		
Shu et al. [77]	2183-Her4 and 888-mel melanoma cell lines	N/A	N/A	Exo-spin columns from cell Guidance systems	N/A	N/A	N/A
Wang et al. [81]	human blood serum	1 × 30 cm	Superose 6 Increase 10/300 GL column	prepacked (GE healthcare)	0.5 mL	AKTA FPLC	yes
Arroyo et al. [83]	Human blood serum and plasma	0.9 × 30 cm	Sephacryl S-500	in-house	25 of 1 mL	NR	yes
Vickers et al. [84]	Human blood plasma	1 × 30 cm or 3 × 10 cm	Superose 6 HR 10/30 or Sephacryl S-200	prepacked (GE healthcare) or in-house	NR	NR	yes
Böing et al. [88]	Human plasma	1.6 × 6.2 cm	Sepharose CL-2B	in-house	26 of 0.5 mL	gravity	yes
De Menezes-Neto et al. [89].	Human plasma	10 mL	Sepharose CL-2B	in-house	30 of 0.5 mL	gravity	yes
Welton et al. [90]	Du145 cell line and human plasma	NR	Exo-Spin™ Midi Columns	Prepacked (CellGS)	30 of 0.5 mL	gravity	yes
Baranyai et al. [91]	Human blood plasma	10 mL	Sepharose 2B, CL-4B, or Sephacryl S-400	in-house	10 of 1 mL	gravity	no
	Rat plasma	120 mL	Sephacryl S-400 (GE Healthcare)	NR	2 mL–5 mL	ÄKTA pure 25 L	yes
Hong et al. [93]	Human plasma	1.5 ×12 cm (10 mL)	Sepharose 2B	in-house	5 of 1 mL	gravity	No
Gámez-Valero et al. [94]	Human blood plasma	12 mL	Sepharose CL-2B	in-house	20 of 0.5 mL	gravity	yes
Jones et al. [106]	Human seminal plasma	35 mL	Sephacryl S300	NR	28 NR volume	NR	No
Aalberts et al. [107]	Sucrose density gradient isolated prostasomes	70 × 2.6 cm	Sephacryl S-1000	prepacked (GE healthcare)	2 mL	NR	No
Brouwers et al. [108]	Human seminal plasma	70 × 2.6 cm	Sephacryl S-1000	prepacked (GE healthcare)	2 mL	NR	No
Lyu et al. [109]	Human seminal plasma	22 × 1 cm	Sephadex G-50 fine	in-house	60 of 0.2 mL	gravity	yes
Kaddour et al. [110]	Human seminal plasma	100 × 1 cm	Gradient of 6 Sephadex beads	in-house	~800 of 0.2 mL	PPLC	yes
Rood et al. [118]	Human urine	30 × 0.78 cm	BioSep-SEC-S4000	prepacked (Phenomenex)	1 mL	Ultimate^TM^ 3000 HPLC	yes
Lozano-Ramos et al. [119]	Human urine	10 mL	Sepharose-CL2B	in-house	20 of 0.5 mL	gravity	yes
Lozano-Ramos et al. [120]	Human urine	10 mL	Sepharose-CL2B	in-house	20 of 0.5 mL	gravity	yes
Ogawa et al. [121]	Human saliva	1.5 × 50 cm	Sepharose-CL4B	in-house	80 NR volume	NR	yes
Ogawa et al. [122]	Human saliva	1.5 × 50 cm	Sephacryl S-500	in-house	100 NR volume	NR	yes
Aqrawi et al. [123]	Human saliva and tears	NR	qEV	prepacked (Izon)	16 of 0.5 mL	gravity	no
Blans et al. [124]	Human and bovine milk	2.5 × 88 cm (432 mL)	Sephacryl S-500	NR	193 of 4.4 mL	FPLC	yes
		238 mL	Sephacryl S-500	NR	119 of 2.65 mL	FPLC	yes
Tietje et al. [125]	Human CSF	N/A	Sephacryl S-400 HR	prepacked spin columns (GE healthcare)	N/A	N/A	N/A
Carreras-Planella et al. [126]	peritoneal dialysis efflux	12 mL	Sepharose-CL2B	in-house	20 of 0.5 mL	gravity	yes
Foers et al. [127]	synovial fluid	2.6 × 60 cm (120 mL)	HiPrep 26/60 Sephacryl S-500 HR	prepacked (GE Healthcare)	NR	NR	yes

NR, not reported; N/A, not applicable; CSF, cerebrospinal fluid; FPLC, fast purification liquid chromatography; HPLC, high performance liquid chromatography; PPLC, particle purification liquid chromatography.

**Table 3 viruses-13-02272-t003:** Summary of EV separation challenges and possible solutions using a PPLC system.

Current EV Purification Challenges		PPLC Solutions
Subpopulation separation requires antibodies whose elution compromises EV integrity	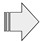	Dye-free separation of intact EV subpopulations
Low reproducibility between different laboratories	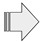	A fully automated fractionation and characterization instrument to reduce handling error
Sample variability (different cell lines, different body fluids, different donors, disease states, etc.)	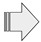	A separation method that is tailored to and valid for each sample type
Characterization performed post-purification introducing handling errors (storage time, temperature, freeze-thaw cycles, etc.)	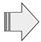	in situ monitoring for real-time complete biophysical characterization
small number of isolated fractions in density gradient purification decreases the power of subpopulation separation	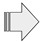	Up to a few thousands of fractions for near single-vesicle resolution
Techniques are either laborious, expensive, and/or require special skills	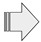	User-friendly, cost-effective system that requires minimum handling effort
Lack of significant room for improvement and scalability in available techniques (density gradient, AF4, immunocapture, etc.)	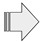	Ample room for improvement by optional addition of extra detectors (light scattering and/or fluorescence) and extra column sizes/chemistries.

## Data Availability

Not applicable.

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
