# Peer review of "The Past, the Present, and the Future of the Size Exclusion Chromatography in Extracellular Vesicles Separation"

_viruses, 2021, doi:10.3390/v13112272_

Round 1

Reviewer 1 Report

The past, the present, and the future of the size exclusion chromatography in extracellular vesicles separation by Hussein Kaddour el al.

This is an excellent review of the extensive literature regarding EV purification with different methods. The authors rightfully cite even the very first groundbreaking papers leading to the discovery of EVs. They present compelling evidence that size exclusion chromatography (SEC) may be the most promising technique for the purification, characterization, and collection of EV subpopulations from biofluids. Particle Purification Liquid Chromatography (PPLC) and hybrid columns on a PPLC platform may be new avenues for EV subpopulation separation with higher resolution along with better EV characterization.

The manuscript is very well written; it was a pleasure to read it. The tables and the figures are excellent. I find it suitable to be published in its present form.

I have no other comments or concerns.

Reviewer 2 Report

In the Manuscript by Kaddour et al., the authors reviewed recent literature on the use of Size Exclusion Chromatography (SEC) to purify Extracellular Vesicles (EVs) and/or to isolate subpopulations thereof. They introduced the general field of EVs and SEC with an historical overview then they illustrated the current methods of EVs purification by SEC from different starting materials (i.e., cell supernatants, blood plasma/serum, semen, urine, and other bodily fluids), and discussed pros and cons and future directions for SEC in the EVs purification field.

The review is without any doubt well organized, structured and written and the topic is thoroughly analyzed. Some parts were approached with a too large technicality (for ex. all the type of resin used for SEC), but I agree with the authors these “technical details” already limit the field and must be underlined.

From the “scientific” point of view the review deserve publication and I have only some small remarks to do:

  1. Line 254: “[97], This…” should be changed into “[97], this…”
  2. Line 325: It seems there is a double space after “qEV columns…”
  3. Lines 381-82: after “bead size,” there is an unnecessary newline

Anyway, beside these minor criticisms I have two main doubts about this MS:

  1. As the MS treat only in limited way aspects connected to viruses (i.e. SEC applied to EVs isolation from HIV patients) I’m guessing why this MS was submitted to “Viruses” (and in the Animal Viruses section) and if the topic matched the journal scope.
  2. Even if the authors clearly stated in a separate section (section 7, Patents) and in the relevant section “Conflict of Interests” that they filed a Patent Application covering the PPLC platform, I’m guessing on the opportunity to publish a review that strongly sustain this method based only on a recent paper published by the same group.

For these reasons I consider the MS suitable for publication from the “scientific” point of view, but, in the light of the above, I decline any final decision on the MS and I leave the final decision to the “Viruses” Editor.
